# Bending Load-Carrying Capacity of Reinforced Concrete Beams Subjected to Premature Failure

**DOI:** 10.3390/ma12193085

**Published:** 2019-09-21

**Authors:** Paolo Foraboschi

**Affiliations:** Dipartimento di Culture del Progetto, Università IUAV di Venezia–Dorsoduro 2206, 30123 Venezia, Italy; paofor@iuav.it

**Keywords:** inelastic analysis, plastic analysis, premature mechanism, rotational capacity

## Abstract

This paper investigates the ultimate flexural strength of reinforced concrete beams when affected by premature failure due to a rotational capacity of the first plastic hinge being consumed before the last plastic hinges reach their maximum possible moment. The paper provides a simple formula for predicting the ultimate load of a hyperstatically supported beam, taking into account the available ductility. The proposed formula is the result of calibration against the ultimate loads from a non-linear analysis on a variety of beams, with a wide spectrum of configurations and with concrete grades from 10.0 to 60.0 N/mm^2^. The formula in based on the plastic hinge model, making it easy to apply, and the ultimate bending moments allow for the actual rotational capacity, making predictions accurate.

## 1. Introduction

Cast in situ reinforced concrete (RC) frames are usually hyperstatic (statically indeterminate), since the ends of a beam are attached to the adjacent columns with moment-resisting connections (beam and column bars passing through the connection).

The simplest way to predict the ultimate load of a statically indeterminate RC beam is using plastic analysis [1,2,3,4]. Accordingly, the beam is envisioned as the composition of three plastic hinges and two rigid elements. Two plastic hinges are at the ends of the beam, while the location of the third one depends on the load. The plastic hinges, hence, connect the two rigid elements to one another and to the boundary columns (two rigid elements between three plastic hinges) [5,6,7].

Plastic analysis allows the ultimate load to be predicted by simple formulas, i.e., using formulas derived from the equilibrium only. Then again, the real ultimate load depends on the actual curvatures that the beam can reach, which, in turn, depend on the inelastic rotational capacity of the plastic hinges, i.e., on the ductility of the cross-sections. Yet, plastic analysis ignores such dependences and makes use of the maximum bending moment that each plastic hinge can reach when it acts individually [8,9,10].

As a result, accuracy of the ultimate load predicted by plastic analysis depends on the differences between the actual bending moment exhibited by each plastic hinge at the ultimate and the maximum bending moment that the plastic hinge can reach without being an element of the beam (i.e., peak of the moment–rotation relationship of the plastic hinge as an individual element) [11,12,13]. Accordingly, plastic analysis may overestimate the ultimate load; a lower ductility leads to greater overestimation [14,15,16].

There are methods to verify that, under the relevant combination of actions, the calculated rotation is less than (or equal to) to allowable plastic rotation [17,18,19]. However, those methods only verify that the rotation capacity is adequate. Ultimately, not only are those methods excessively complicated, but they also do not predict the ultimate load for premature failure.

The more accurate way to predict the ultimate load of a statically indeterminate RC beam is using non-linear analysis, which allows inelastic rotational capacity to be taken into account in the predictions of the ultimate load [20,21,22,23].

A non-linear analysis of a RC structure is not simple. However, it can be performed using the finite element method, which not only simplifies the work but also allows complex models to be created [3,6,10,24,25]. In so doing, the structure can be analyzed in its entirety—namely, modeling all the RC columns and beams that compose the frame [26,27]. However, non-linear analysis is cumbersome even if performed using a software package and even if restricted to just the RC beam. Moreover, in common structural engineering software, the data are laborious to input, and the output is complicated to read, especially in the case of existing buildings. Hence, the finite element method can be used only when the design is completed, in order to perform the code verifications, while it is not appropriate at the design stage to obtain the optimal structure. Thus, non-linear analysis is not a design tool, but is a tool for final code verifications in the most complex situations.

Non-linear analysis should also take into account the crack pattern, tension stiffening effect, creep, shrinkage, and concrete strength both in tension and compression. Recent significant publications relating to those issues are References [28,29,30,31,32,33,34,35,36,37,38,39,40]. The finite element method implemented in advanced and sophisticated structural analysis programs allows those phenomena to be accounted for, as long as several parameters are plugged into the finite element model. In the case of existing structures, however, those parameters cannot be known accurately. Thus, on one hand, a non-linear analysis is the most accurate way of modeling a RC structure, both new and existing. On the other hand, however, the degree of accuracy that can be achieved by a non-linear analysis in the case of an existing structure is definitely not commensurate with the effort and time that are required to run the analysis.

As a natural consequence, practitioners are used to performing strength verification through either linear elastic analysis or linear elastic analysis with redistribution [9,41]. The former uses the maximum elastic bending moment, while the latter uses a ratio of the elastic bending moment, which is called “redistributed moment”. However, both of the methods assume that failure is dictated only by the section that reaches the maximum bending moment (either elastic or redistributed), while the contributions provided by the other sections are ignored. The in-field consequence is that common engineering structural practice disregards the contributions of the sections that become plastic hinges apart from the first plastic hinge.

The main shortcoming of calculating the ultimate load with a linear elastic analysis (with or without redistribution), as well as by a plastic analysis, is that the formulas do not adequately recognize the relationship between the height of the concrete section and the amount of longitudinal steel reinforcement. In fact, according to those formulas, a reduction in height of the cross-section can be compensated by an increase in the amount of reinforcement, while in reality this is true only if the amount of reinforcement is not large.

The motivation for this study was to obtain a simple formula that predicts the ultimate bending load of an RC beam, along with the accuracy of non-linear analysis and the straightforwardness of plastic analysis. Such a formula is proposed for both the design stage and safety verification purpose.

## 2. Study’s Statement of Purpose, Terminology, and Framing the Research Question

This paper deals with RC beams (i.e., RC structural elements carrying transverse external loads that cause bending moment and shear force along their length), which are hyperstatically supported at the ends (i.e., each beam is a part of a frame or of a multi-span beam).

The paper focuses on bending failure, while shear failure is not considered herein. Specifically, the transverse reinforcement provides the reference structure with a shear strength greater than the bending strength. Since an RC beam can also collapse by shear failure, design and safety assessment must supplement the prediction of the bending load-carrying capacity with the prediction of the shear load-carrying capacity of the RC beam [42,43,44,45].

Accordingly, the mechanism that dictates the ultimate load of the RC beam is composed of three plastic hinges (two at the ends and one along the span), whose development occurs one following the other with a considerable gap. Due to that gap, the rotational capacity of the first plastic hinge can be lower than that required to form all the plastic hinges of the failure mechanism. In that case, the load-carrying capacity stems from a premature failure mode.

### 2.1. Terminology Used in the Paper

In the interest of clarity, this paper uses the terminology described below.

The “moment capacity”, which is denoted by *M_max_*, is the peak ordinate of the moment–curvature relationship of an individual element of the beam, separated and independent from the beam. The abscissa of that ordinate is referred to as “curvature capacity” and is denoted by χ*_max_*. The element of the beam that is referred to herein is the plastic hinge. Thus, the moment capacity and curvature capacity that are referred to hereinafter are those of the plastic hinges—i.e., the individual capacity of each plastic hinge when acting as an isolated RC element. Hence, *M_max_* is the individual maximum (bending) moment.

The “ultimate moment”, which is denoted by *M_u_*, is the bending moment acting on a plastic hinge when the beam reaches the ultimate, i.e., when the beam furnishes the load-carrying capacity. Thus, the ultimate moment that is referred to hereinafter is the maximum bending moment reachable by a plastic hinge when acting as a part of a whole RC beam. Hence, *M_max_* is the (bending) moment of a plastic hinge at the ultimate of the beam.

By definition, *M_u_* ≤ *M_max_*. The greater the difference is between the curvature capacity and the actual curvature of a plastic hinge when the beam is at ultimate, the greater the difference between *M_max_* and *M_u_* will be.

A “fully developed failure mode” is a collapse mechanism whose plastic hinges are carrying their individual maximum moment *M_max_* while undergoing the rotations induced by the mechanism. Hence, the ultimate moment of every plastic hinge of a fully developed failure mode is *M_u_* = *M_max_*.

A “premature failure mode” is a collapse mechanism whose plastic hinges are not all carrying their individual maximum moment while undergoing the rotations induced by the mechanism. Hence, the ultimate moment of one or two plastic hinges of a premature failure mode is *M_u_* < *M_max_*.

The ultimate load is the maximum load that may be placed on the beam before its failure. This paper focuses on the ultimate load associated with the bending failure mode. Thus, the ultimate load that is referred to is the bending load-carrying capacity of the RC beam (i.e., they are synonyms).

### 2.2. Theoretical Framework

The ultimate load of a fully developed failure mode is balanced by the moment capacity *M_max_* of all three plastic hinges of the beam. The ultimate load of a premature failure mode is balanced by the moment capacity *M_max_* of one or two plastic hinges of the beam and by the ultimate moment *M_u_* < *M_max_* of two or one plastic hinges.

All other things held constant, if the beam fails by the fully developed failure mode, the ultimate load is greater than if the beam fails by a premature failure mode. That is, all the rest being equal, the fully developed failure mode is the collapse mechanism that provides the beam with the maximum (bending) load-carrying capacity, while a more premature failure mode results in a lower (bending) load-carrying capacity of the RC beam.

The difference between the load-carrying capacity that a beam would have if it collapsed by a fully developed failure mode and the actual load-carrying capacity of the beam that collapses by a premature failure mode is herein dealt with as potentiality not actualized. Specifically, the former is the potential capacity and the latter is the capacity that can be actualized.

A fully developed failure mode requires high curvature capacity of the first plastic hinges that develop, so as to allow the last plastic hinge that develops to reach its moment capacity. That requirement needs ductile plastic hinges. Thus, a premature failure mode is the result of one or more plastic hinges with little ductility. The ductility is, hence, the property that allows the potentiality in terms of load-carrying capacity to be actualized in reality. Therefore, a greater ductility of the beam results in a closer *M_u_* to *M_max_* for the plastic hinges.

The problem being addressed in this paper is the prediction of the premature failure mode due to lack of ductility, allowing for the actual bending moments *M_u_* that determines the load-carrying capacity.

The load-carrying capacity of a beam can be predicted accurately and easily, insofar the bending moments *M_u_* are predicted accurately and easily. If the beam actualizes its full potentiality, *M_u_* = *M_max_* for every plastic hinge. The moment capacity *M_max_* can be predicted using models that are not only simple but also reliable, as proven by decades of successful use. Thus, the load-carrying capacity of a beam subjected to fully developed failure mode can be predicted accurately and easily.

If the beam does not actualize its full potentiality, *M_u_* < *M_max_* for one or two plastic hinges. In that case, the moment at the ultimate must be predicted by modeling the rotational capacity of the entire beam. However, such models are accurate only if they are complex. Therefore, the load-carrying capacity of a beam subjected to premature failure mode can be predicted either accurately but laboriously, or easily but approximated.

Ultimately, until now, accuracy and convenience were always in conflict when predicting the ultimate load of RC beams. A main aim of the present research work was to bridge the above-described gap between accuracy and convenience in predicting the strength of RC beams. That goal was reached by using a method specifically developed within the framework of plastic analysis, whereas conventional plastic analysis disregards premature failure modes. More specifically, the formula of plastic analysis was tuned so as to take into account the rotational capacity that is disregarded by the basic plastic analysis, which led to a plain formula.

This paper gives a detailed account of the formula that was obtained for predicting the flexural load capacity of hyperstatic RC beams, including its derivation and its application. That formula expresses the percentage of load-carrying capacity that is in potentiality but that cannot actualize because of a lack of inelastic rotational capacity, i.e., insufficient ductility available. In so doing, the ultimate load allows for the actual ductility of the beam, making it accurate, and it is also easy to calculate, since it stems from an equilibrium equation.

## 3. Reference Structure

The diagram of the reference structure in a hyperstatically supported RC beam is shown in the schematic of Figure 1, including the main symbols. In particular, the effective depth of the cross-section is denoted by *d*, and the depth of the neutral axis is denoted by *y*. Accordingly, the ratio of the neutral axis depth to effective depth is denoted by *y*/*d*. It is to note that this paper used the ratio of the neutral axis depth to effective depth *y*/*d* at the ultimate—i.e., the depth of the neutral axis *y* that is referred to is when the cross-section transmits the ultimate bending moment.

The paper focuses on the uniform load *q*, and the formula that is presented provides its ultimate value, *q_u_*.

Nevertheless, the results of the theoretical analysis carried out in this research hold true for any type of loadings. Therefore, the formula can be easily rearranged in order to predict the ultimate value of other loading distributions, including concentrated forces.

RC beams confined by means of adequate closed stirrups or cross-ties are not subjected to premature failure, since the transverse reinforcement confines the concrete and, therefore, increases the ductility of the beam. Premature bending failure may occur only if the stirrups provide the RC beam with adequate shear load-carrying capacity but no significant confinement. Ultimately, confinement effects are not considered, because confinement would guarantee a fully developed failure mode. Accordingly, the bending load-carrying capacity that is predicted does not depend on the stirrups, which are herein ignored.

The focus of the research—the difference between potential and real bending load-carrying capacity—suggests introducing the following ratio μ:(1)$$\mu =\frac{q_{u}}{q_{ud}}$$
in which *q_u_* is the real ultimate load of the reference structure (load-carrying capacity for the actual flexural failure mode) and *q_ud_* is the ultimate load that the reference structure would have if, at the ultimate, the plastic hinges all simultaneously reached the moment capacity *M_max_* (bending load-carrying capacity for the fully developed mode).

Obviously, μ ≤ 1. A lower μ results in a more premature failure. If μ = 1, the beam collapses by a fully developed failure mode. In this case, the beam actualizes all potentiality regarding the bending load-carrying capacity. Hence, μ expresses the fraction of the potential that the RC beam can actualize. It is worthy specifying that μ can be equal to one not only in confined beams, but also in unconfined beams whose span-to-depth ratio and longitudinal reinforcement amount are not extreme or excessive. That is, the ductility of unconfined RC beams can be enough for having a fully developed failure mode.

## 4. Model

The results against which the novel formula was calibrated were derived from a non-linear analysis of an exhaustive variety of RC beams, which aimed to supply accurate predictions of the ultimate load, along with stresses, strains, curvatures, and displacements. Those predictions were obtained using an analytical model specifically developed for RC beams, which simulated their behavior up to failure. That analytical model is based on the assumptions, constitutive relations, and methods described in this section. The references highlight advances at the cutting edge of research relative to non-linear modeling of RC structures [46,47,48,49,50,51,52].

The basic assumption of the model is that plane cross-sections remain plane. Accordingly, the deformation of the beam can be analyzed using the curvature χ of the cross-sections. The model is, hence, based on the function χ(*x*), where *x* is the axis that identifies the cross-sections (Figure 1).

As mentioned in Section 3, the concrete is unconfined, since, when it is properly confined, the rotational capacity of the first plastic hinge is not consumed prior to the triggering of the remaining plastic hinges; thus, the confined beam is not subjected to premature failure. Therefore, the stress state is uniaxial.

The stress–strain relationships of concrete for uniaxial compression and tension due to short-term loading is modeled using functions that accurately describe the main properties of the ascending and descending parts [53]. Compressive stresses and strains are herein assumed to be positive.

The relationship between the compression stress σ*_c_*, in N/mm^2^, and the compression strain ε*_c_* is described by the following function, which is in the form shown schematically in Figure 2:(2)$$\sigma_{c}=\frac{\frac{E_{c}}{E_{c1}}\cdot \frac{\varepsilon_{c}}{\varepsilon_{c1}}-{\left(\frac{\varepsilon_{c}}{\varepsilon_{c1}}\right)}^{2}}{1+\left(\frac{E_{c}}{E_{c1}}-2\right)\cdot \frac{\varepsilon_{c}}{\varepsilon_{c1}}}\cdot f_{cm}$$
where the stress *f_cm_* is the mean value of the concrete compressive strength, expressed in N/mm^2^, ε*_c1_* = 2.2‰, *E_c_*_1_ = *f_cm_*/ε_c1_, and *E_c_* is the tangent modulus according to Equation (3), in N/mm^2^.
(3)$$E_{c}=22000\cdot {\left(\frac{f_{cm}}{10}\right)}^{0.3}$$

For the descending part of the stress–strain function, Equation (2) is valid only for values of σ_c_/*f_cm_* ≤ 0.5. The strain ε_cL_ at σ_cL_ = 0.5·*f_cm_* may be calculated from Equation (4).
(4)$$\varepsilon_{cL}=\varepsilon_{c1}\cdot \left\{\frac{1}{2}\cdot \left(\frac{1}{2}\cdot \frac{E_{c}}{E_{c1}}+1\right)+{\left[\frac{1}{4}\cdot {\left(\frac{1}{2}\cdot \frac{\;E_{c}\;}{\;E_{c1}\;}+1\right)}^{2}-\frac{1}{2}\right]}^{0.5}\right\}$$

For strains ε*_c_* > ε*_cL_* the descending branch of the σ*_c_*–ε*_c_* function is described using Equations (5)–(7).
(5)$$\sigma_{c}=f_{cm}\cdot {\left[\left(\frac{\zeta }{\lambda }-\frac{2}{\lambda^{2}}\right)\cdot {\left(\frac{\varepsilon_{c}}{\varepsilon_{c1}}\right)}^{2}+\left(\frac{4}{\lambda }-\zeta \right)\cdot \frac{\varepsilon_{c}}{\varepsilon_{c1}}\right]}^{-1}$$
with
(6)$$\zeta =\frac{4\cdot {\left[\lambda^{2}\cdot \left(\frac{E_{c}}{E_{c1}}-2\right)+2\cdot \lambda -\frac{E_{c}}{E_{c1}}\right]}_{}}{{\left[\lambda \cdot \left(\frac{E_{c}}{E_{c1}}-2\right)+1\right]}^{2^{}}}$$
and with
(7)$$\lambda =\frac{\varepsilon_{cL}}{\varepsilon_{c1}}$$

Compressive failure of concrete is often a discrete phenomenon, i.e., there is a fracture region of limited width, in which compression strains are concentrated. For practical reasons and due to lack of sufficient experimental data, these strain concentrations are generally smeared as done in Equations (1)–(6). As a consequence, the descending branch of the stress–strain relationship in compression is influenced by the length of the member subjected to compression.

Due to such uncertainty and highly variable performance, the descending portion of the stress–strain relationship is considered as an envelope to all possible stress–strain relationships of concrete, which tends to soften as a consequence of concrete micro-cracking. In view of that, the concrete constitutive law of Equations (4) and (5) can be used either imposing a limit on ε*_c_* or without imposing a limit on ε*_c_*. That limit is the crushing strain of concrete, which is herein denoted by ε*_cu_*. In the former case, the contribution to the bending moment provided by the concrete compression force depends on the crushing strain ε*_cu_*, while, in the latter case, it does not depend on ε*_cu_*.

The results of the non-linear analysis demonstrated that the contribution to the bending moment provided by the concrete compression force depends slightly on ε*_cu_*, since a greater ε*_cu_* suggests a lower lever arm of the concrete compression force, and vice versa (i.e., the greater the strain reached at the concrete compression edge is, the closer the position of the concrete compression force is to the position of the steel tensile force). As a result, the load-carrying capacity was found to depend only marginally on ε*_cu_*. On that account, Equations (4) and (5) are herein used without any limit on ε*_c_*, i.e., without applying ε*_cu_*.

Tensile failure of concrete is always a discrete phenomenon, as well as compressive failure. Unlike compressive failure, tensile failure of concrete was extensively studied by fracture mechanics and the results obtained allow cracks to be accurately modeled. Furthermore, smearing the strain concentrations would yield lower accuracy than for compressive failure.

Thus, to describe the tensile behavior a stress–strain diagram is here used for the uncracked concrete, and a stress–crack opening diagram is used for the cracked section. More specifically, for uncracked concrete subjected to tension, a bilinear stress–strain relationship is used, which is composed of two ascending branches with a decreasing slope. For a cracked section, a bilinear stress–crack opening relationship is used, which is composed of two descending branches with a decreasing negative slope.

The relationship between the stress σ*_s_* and the strain ε*_s_* of the reinforcing steel is described by the elasto-plastic function, assuming a modulus of elasticity *E_s_* equal to 210,000 N/mm^2^, both in tension and compression.

The compressive and tensile stress–strain relationships and the stress–crack opening relationship of concrete, and the stress–strain relationship of reinforcing steel were merged in the moment–curvature relationship. Thus, the model deals directly with the moment–curvature relationships, while the other relationships are used only indirectly.

### Moment–Curvature Relationship

The deformation of the RC beam is calculated using the relationship between moment and curvature, *M*–χ, described in this sub-section. As already mentioned, the *M*–χ relationship is derived from the analytical stress–strain relationships of concrete and reinforcing steel described above.

At the ultimate, the beam is cracked, and cracking is in the stabilized crack pattern [53]. In a cracked section, all tensile forces are balanced by the steel only. However, between adjacent cracks, tensile forces are transmitted from the steel to the surrounding concrete by bond forces. The contribution of concrete increases, therefore, the stiffness of the tensile reinforcement (the well-known tension stiffening effect).

The model takes the tension stiffening effect into account by a modified moment–curvature relationship that uses the mean curvature χ instead of the actual curvature. The mean curvature χ at any section of a beam is determined as shown below [53].

For uncracked state:(8)$$\chi =\chi_{1}$$

For stabilized cracking state:(9)$$\chi =\chi_{2}-0.80\cdot \left(\chi_{cr2}-{\;\chi }_{cr1}\right)\cdot \frac{M_{cr}}{M}$$

For post-yielding state:(10)$$\chi =\chi_{y}-0.80\cdot \left(\chi_{cr2}-{\;\chi }_{cr1}\right)\cdot \frac{M_{cr}}{M_{y}}+\frac{M-M_{y}}{2\cdot \left(M_{max}-M_{y}\right)}\cdot \left(\chi_{max}-\chi_{y}\right)$$

The nomenclature used in Equations (8)–(10) is as follows: *M* is the acting bending moment; *M_y_* is the yielding moment; *M_max_* is the moment capacity defined in Section 2; *M_cr_* is the cracking moment; χ*_y_* and χ*_max_* are the curvature corresponding to *M_y_* and *M_max_*, respectively (i.e., χ*_max_* is the curvature capacity defined in Section 2); χ_1_ and χ*_cr_*_1_ are the curvature in uncracked state corresponding to the actions *M* and *M_cr_*, respectively; χ_2_ and χ*_cr_*_2_ are the curvature in the naked stabilized cracking state corresponding to the actions *M* and *M_cr_*, respectively.

The cracking moment *M_cr_* is determined as follows [53]:(11)$$M_{cr}=W_{1}\cdot f_{ctd}$$
where *W*_1_ is the section modulus in uncracked state including the reinforcement, and *f_ctd_* is the design value of concrete tensile strength.

The diagram of the *M*–χ relationship described by Equations (8)–(10) is well known (Figure 3). The curve is composed of four branches—namely, an early linear ascending branch, which starts from zero; a second horizontal branch; a third curved ascending branch, with increasing slope; and a fourth straight ascending branch up to the moment capacity. Hence, the curve does not exhibit any descending branch.

The *M*–χ relationship allows the deformation of the beam to be computed using the sections of the beam, while the rotational capacity and the length of the plastic hinges are used only indirectly (i.e., the rotational capacity and the length of the plastic hinges are implicit within the calculation of the deformation).

In general, the moment capacity *M_max_* of a section (of an element of the beam) can be defined either as the maximum moment that the section can transmit or as the moment at concrete failure. Since this study was carried out without any limit on ε*_c_*, as previously mentioned, *M_max_* is herein the maximum bending moment of the section (i.e., of the element of the beam), and as such is dictated by the yielding of the tension steel.

The yielding moment *M_y_* and the moment capacity *M_max_* of Equations (8)–(10) are calculated using Equations (2)–(7) (Figure 2), together with the constitutive law of the reinforcing steel. The greater the tension reinforcement ratio is, the lower the difference between *M_max_* and *M_y_* will be, whereas the former is always greater than the latter.

## 5. Method

The models described in Section 4 allow the ultimate load to be predicted in the framework of a non-linear analysis, accounting for the actual bending moment *M_u_* that each section of the RC beam reaches at the ultimate, which may be different from the maximum bending moment *M_max_* that each element could reach at the ultimate (maximum moment) acting individually (see Section 2 for the terminology).

The method arranged to calculate the ultimate load of a RC beam with given geometry is described below (Figure 4).

The first step of the method is to calculate the *M*–χ relationship for the section at midspan and for the sections at the ends of the beam, as well as for all the sections whose geometry (concrete or reinforcement) is different than the geometry of the ends or midspan sections.

The second step refers to the moment capacity *M_max_* calculated at the first step for the sections at the left end and right end of the beam (*M_max_* is the last point of the *M*–χ relationship). Those moment capacities are denoted by $M_{max}^{l}$ and $M_{max}^{r}$, respectively.

The moments $M_{max}^{l}$ and $M_{max}^{r}$ are applied at the ends of the beam, together with a uniform load *q* and the shear forces *V^l^* and *V^r^* (Figure 4). The load *q* is unknown for the meantime. The shear forces *V^l^* and *V^r^* derive from the equilibrium. Thus, while $M_{max}^{l}$ and $M_{max}^{r}$ are given (they only depend on the relevant end section), *V^l^* and *V^r^* depend on *q* (which for the time being is unknown) and on the difference between $M_{max}^{l}$ and $M_{max}^{r}$.

Then, the beam is analyzed without restraints, as the end forces are known (Figure 4). In so doing, the beam can be dealt with as statically determinate, under the provisional assumption that the bending moments that result in the other sections of the beam from equilibrium are lower than (or equal to) the moment capacity *M_max_* of the relevant sections. That aspect is checked and, if necessary corrected in the last two steps of the procedure.

This step ends with the calculus of the load *q* that produces nil relative rotation between the sections at *x* = 0 and *x* = *L*. That calculation is expressed by the following equation:(12)$$\int_{0}^{L}\chi \left(q,x\right)dx=0$$

Equation (12) is the compatibility equation, which expresses that the integration of the curvatures between the fixed ends must be zero, where the curvature depends on *x* and *q*.

The curvature χ within the definite integral is unknown. Nevertheless, χ can be expressed as a function of the moment *M* acting on each section of the beam, which is known from equilibrium, as the actions at the ends of the beam are known (Figure 4). In so doing, the function χ that is integrated is known, barring *q*. Thus, the definite integral of Equation (12) can be expressed as a function of *q*. Then, the left member of Equation (12) can be equated to zero, which allows the transverse load *q* to be obtained.

It is to note that Equation (12) has one and only one solution, because the *M*(χ) relationship (Section 4) is an injective and surjective function (it is composed of ascending branches only). If the *M*(χ) relationship had a descending branch, not only would Equation (12) have many solutions, but the actual solution would also be numerically difficult to achieve.

From the operational point of view, the load *q* that produces nil relative rotation between the ends of the beam can be determined using the *M*–χ relationships of the beam sections, starting from an initial value of *q*, which is increased (or decreased) step by step up to finding the value of *q* that verifies Equation (12). That value of the load is the provisional value of *q_u_*.

The third step of the method is devoted to verifying that every section of the RC beam (in particular, the sections around midspan) can exhibit the relevant curvature used in Equation (12). Specifically, this step checks that each section of the beam can transmit the bending moment considered in the left member of Equation (12) in order to calculate the relative rotation between the beam’s ends.

If the value of *M* in every section of the beam is lower than the relevant moment capacity *M_max_*, the load calculated at the second step is the actual ultimate load *q_u_*. In other words, the provisional value of *q_u_* becomes definitive. If, conversely, that condition is not verified, the procedure goes on with a fourth step.

The fourth step uses the moment capacity in the section where the bending moment used for Equation (12) at the second step exceeds the moment capacity (i.e., in the section where *M* surpasses *M_max_* most). That section is at, or around, midspan. That moment capacity is denoted by $M_{max}^{m}$.

A plastic hinge is then placed at that section. In more detail, the section along the span that at the second step exceeded the relevant moment capacity is transformed into a pin with two moments equal to $M_{max}^{m}$ applied at the two sections hinged by the pin. After that, the second step is repeated for the beam with that plastic hinge along the span.

From the operational point of view, this new second step starts with an initial value of *q*, which is increased (or decreased) step by step up to finding the value of *q* that verifies Equation (12), using the *M*–χ relationships of the beam sections. However, in this new step two, the equilibrium is used to determine not only the shear forces *V^l^* and *V^r^*, but also the ultimate moments at the left and right ends $M_{u}^{l}$ and $M_{u}^{r}$, respectively, which are lower than $M_{max}^{l}$ and $M_{max}^{r}$. In doing so, the beam can be analyzed without restraints (statically determinate), as already done at the second step. The novel value of *q* that verifies Equation (12) is the actual ultimate load *q_u_* of the RC beam.

## 6. Non-Linear Analysis

The formula for the ultimate load that this research aimed at attaining was derived based on a theoretical non-linear analysis devoted to calculating the ultimate load of a variety of beams with a large spectrum of configurations, so as to consider all the possible rotational capacities and ductilities of real RC beams. The non-linear analysis was carried out using the method described in Section 5, which in turn uses the model described in Section 4.

Many RC beams were analyzed, each different in span, cross-section, concrete grade, and reinforcement arrangement. More specifically, the set of beams chosen to describe all possible practical applications was composed of rectangular and T sections, of height (depth) from 150 mm to 1500 mm, and width from 150 mm to 1300 mm. Each section was analyzed with low-to-high longitudinal reinforcement ratios—i.e., the analysis included low-to-high ratios of the area of the tension reinforcement to the area of the concrete and low-to-high ratios of the area of the compression reinforcement to the area of the concrete, with almost all the combinations. Moreover, each beam was analyzed with a concrete grade that ranged from *f_cm_* = 10.0 N/m^2^ to *f_cm_* = 60.0 N/m^2^ (see Equation (2) for the symbols). The steel reinforcement that was considered had a characteristic value of the tensile strength equal to 450.0 N/mm^2^ and a modulus of elasticity of 210,000 N/mm^2^ both in tension and compression.

The thickness of the concrete cover was taken as 25 mm. Accordingly, the effective depth was taken as 50 mm less that the depth of the beam, to account for the thickness of the stirrups and the position of the centroid of the tension reinforcement.

Since the theoretical analysis covered a set of RC beams that represented almost all the realistic practical applications, the outcomes of the analysis allowed general results to be derived. The results were examined methodically and in detail. Insight was devoted not only to examining the ultimate loads *q_u_*, but also to explaining and interpreting all the results, as well as to studying closely the nature and relationship of the parts that compose the RC beams. The product of that activity is checked in Section 7 and summarized in Section 8.

## 7. Experimental Verification of the Theoretical Results

The verification of the predictions from the analytical modeling was performed using the existing experimental data found in the literature. To that purpose, the non-linear analysis included some RC beams that were experimented for which test results were available in the literature [54,55,56,57]. Some of those RC beams were tested by the author [58,59,60], within research programs on strengthening of RC beams using external reinforcement (those research programs also included testing the RC beams prior to applying the external reinforcements).

The results from such experiments provided a benchmark against which the results from the non-linear analysis could be verified. Comparisons between theoretical and experimental results were made in terms of ultimate load and ultimate moments. The comparisons showed a good agreement between the results from the non-linear analysis and the results borrowed from the literature.

Table 1 shows some comparisons referring to the ultimate moments reached by the last plastic hinge. Those comparisons used the experimental results that are reported in Reference [61]. Section 10 deals with an RC beam whose experimental behavior is known [60].

## 8. Results Derived From the Outcomes of the Non-Linear Analysis

One of the main results of the non-linear analysis was the ratio μ of the potential load-carrying capacity that the rotational capacity allowed the RC beams to reach (defined by Equation (1) in Section 3).

The fraction μ of each RC beam was expressed as a function of the rotational capacity of the plastic hinges, which, in turn, was expressed in relation to the ratio of the area of the tension reinforcement to the area of the concrete at the critical sections of the beam, the ratio of the area of the compression reinforcement to the area of the concrete at the critical sections, the ratio of the ultimate neutral axis depth to effective depth *y*/*d*, the ratio of the span to the depth, the concrete grade, and the shape of the concrete cross-section. In so doing, several relationships were obtained, which displayed the dependences of μ on the characteristics of the beam.

Inspection revealed that the relationship which governs is *y*/*d* of the plastic hinge that develops first, which dictates the rotational capacity of the first plastic hinge. In turn, that rotational capacity governs the rotation that the last plastic hinge can reach. Eventually, the rotation reached by the last plastic hinge dictates the bending moment reached by the last plastic hinge.

The beams that have the same *y*/*d* ratio for the first plastic hinge also have approximately the same value of μ, even if the tension and compression reinforcement ratios, reinforcement arrangements, span-to-depth ratios, concrete grades, and cross-sectional shape are greatly different from each other. That result allowed the ultimate loads from the non-linear analysis to be expressed as a one-variable function—i.e., a relationship between μ and *y*/*d*.

That function was constructed using the mean ultimate load of the beams whose first plastic hinge that formed had almost the same *y*/*d* ratios. Specifically, the value of the function that was built, associated to each *y*/*d* (abscissa) the mean ultimate load (ordinate) averaging the values obtained for the beams whose first plastic hinge had a *y*/*d* ratio in the range ±1% around the abscissa.

The one-variable function that was defined was used as a benchmark. The formula of the plastic analysis was tuned against that one-variable function.

According to the ultimate strength analysis (see Section 1), the ultimate load *q_u_* is the load that is balanced by the ultimate bending moments *M_u_* in the three plastic hinges that develop in the beam. The ultimate load can be calculated using the virtual work (of course, it can be calculated using the equilibrium directly, but the virtual work is definitely more rapid and more emblematic in this case).

For uniform loads, the hinge along the span can be assumed to be exactly at midspan even if the ultimate moments of the two plastic hinges at the ends are different from one another, since the error that results is always marginal. Thus, if the load is uniform, *q_u_* from ultimate strength analysis is as follows:(13)$$q_{u}=\frac{4\cdot {\left(2\cdot M_{u}^{m}+M_{u}^{l}+M_{u}^{r}\right)}_{}}{L^{2}}$$
where $M_{u}^{m}$, $M_{u}^{l}$, and $M_{u}^{r}$ are the ultimate bending moment in the plastic hinges at midspan, at the left end, and at the right end, respectively, while *L* is the span of the beam.

It is to note that $M_{u}^{l}$ is the minimum between the ultimate bending moment of the left end and of the structure at the left boundary (left restraint), and $M_{u}^{r}$ is the minimum between the ultimate bending moment of the right end and of the structure at the right boundary (right restraint). The left or right restraint may be an adjoining beam or a column (or both of them).

As mentioned several times before, $M_{u}^{m}$, $M_{u}^{l}$, and $M_{u}^{r}$ depend on the whole beam and not only on the plastic hinges acting individually (each moment does not depend on the relevant section only). Replacing those three bending moments with the moment capacity *M_max_* of the three plastic hinges, on one hand, the moments of Equation (13) can be obtained easily from the sectional analysis, but on the other hand, those moments *M_max_* might cause Equation (13) to overrate the prediction of the ultimate load. In fact, the ultimate behavior of the beam may not correspond to the simultaneous attainment of *M_max_* in the three plastic hinges—i.e., the ultimate load may be not balanced by the moment capacity of the three plastic hinges, because the moments acting on one or two plastic hinges may be lower.

The research method pursued in this study was to calibrate the mechanism equation of ultimate strength analysis, i.e., Equation (13). The calibration process consisted of referring the ratio between *M_u_* and *M_max_* of each plastic hinge, and tuning such ratios against the results from the non-linear analysis.

The outcomes from the non-linear analysis demonstrated that the first plastic hinge that forms transmits a bending moment equal to the moment capacity *M_max_* of that plastic hinge, while the last plastic hinge that forms transmits a bending moment that depends on *y*/*d* of the first plastic hinge that formed.

The outcomes also demonstrated that the two plastic hinges at the ends transmit the same fraction of the relevant *M_max_*.

Those results enabled a fresh synthesis to be produced—i.e., calibration could be performed parametrizing only the bending moment of the last plastic hinge (if at midspan) or of the last plastic hinges (if at the ends, which are related to one another). Parametrization consisted of expressing the bending moment (moments) to be tuned as an unknown fraction of the moment capacity *M_max_* of the last plastic hinge (hinges).

For the purpose of calibration, *M_max_* was expressed in a general and basic form, which allowed the mechanism equation to be tuned at best.
(14)$$M_{max}=A_{s}\cdot f_{sd}\cdot \left(d-0.4\cdot y\right)+A_{s}^{\prime }\cdot f_{sd}\cdot \left(0.4\cdot y-c^{\prime }\right)$$
where *A_s_* and $A_{s}^{\prime }$ are the area of the tension and of the compression reinforcement in the plastic hinge that transmits *M_max_*, the stress *f_sd_* is the design strength of the steel reinforcement, and c’ is the distance of the compression reinforcement from the compression edge of the cross-section. The symbol *d* was already defined (Section 3), i.e., it is the effective depth.

Ultimately, *M_max_* provided by Equation (14) is the estimation of the moment capacity that was used for calibrating Equation (13), whereas the formula obtained with the calibration can be used for estimating *M_max_* with a different expression. However, Equation (14) almost always provides good accuracy.

In many cases, the contribution of the compression reinforcement of Equation (14) can be ignored. In so doing, the calculation is simplified, with only a marginal loss of accuracy.

The parametric form consisted of expressing the bending moment(s) to be tuned as the value provided by Equation (14) multiplied by a parameter, which is herein denoted by δ; the former was known while the latter was unknown. The essential feature of δ is the fact that it reduces the moment acting on a plastic hinge. Calibration consisted of reducing the bending moment of the last plastic hinge(s) with respect to the relevant moment capacity *M_max_*, based on the *y*/*d* ratio of the first plastic hinge. The parameter(s) δ was (were) tuned so as to minimize the difference between *q_u_* from the non-linear analysis, whose models and method are described in Section 4, Section 5 and Section 6, and from Equation (13) with the bending moment(s) expressed as the fraction(s) of the relevant *M_max_*. Minimization was accomplished using the least square method.

Activity moved forward with the calculation of the values of δ that allowed the parametrized equation to reproduce the values of *q_u_* from the non-linear analysis.

Then, activity searched for the analytical function of *y*/*d* (at the ultimate) that best fit those values of δ.

The search led to the conclusion below. The bending moment transmitted by the last plastic hinge(s) can be expressed in the following form:*M’_u_* = δ⋅*M_max_*(15)
where δ = 1.0; for *y*/*d* ≤ 0.15,
(16)$$\delta ={e^{{}^{-10\cdot \left(\frac{y}{d}-0.15\right)}}}^{2}\;\mathrm{for}\;y/d\;>\;0.15$$
in which *M’_u_* is the bending moment transmitted by the last plastic hinge(s) at the ultimate, *M_max_* refers to the moment capacity of the last plastic hinge(s) provided by Equation (14), and *y*/d is the ratio of the first plastic hinge(s). The value of δ as a function of the *y*/d ratio is shown in Figure 5.

Ultimately, Equation (15) provides *M’_max_* of the last plastic hinge/hinges that develops/develop, starting from *M_max_* provided by Equation (14) and using δ provided by Equation (16) with the *y*/d ratio of the first plastic hinge. The accuracy of Equations (15) and (16) is proven in Section 10 (Figure 6).

## 9. Predictive Expression

Equations (15) and (16) allow the ultimate load *q_u_* of an RC beam to be predicted allowing for possible premature failure.

If the last plastic hinge forms around midspan, the ultimate load is provided by the following equation:(17)$$q_{u}=\frac{4\cdot {\left(2\cdot \delta \cdot M_{max}^{m}+M_{max}^{l}+M_{max}^{r}\right)}_{}}{{L^{2}}^{}},$$
in which δ is calculated using the *y*/*d* ratio of the end sections (if those sections are different from one another, δ refers to the weakest end section).

If the last plastic hinge forms at one of the ends, the ultimate load is provided by the following equation:(18)$$q_{u}=\frac{4\cdot {\left(2\cdot M_{max}^{m}+\delta \cdot M_{max}^{l}+\delta \cdot M_{max}^{r}\right)}_{}}{{L^{2}}^{}},$$
in which δ is calculated using the *y*/*d* ratio of the midspan section.

The nomenclature of Equations (17) and (18) was defined in Section 5, i.e., $M_{max}^{m}$, $M_{max}^{l}$, and $M_{max}^{r}$ are the moment capacity at midspan, left end, and right end, respectively. Those moments can be calculated using Equation (14).

As previously noticed, $M_{max}^{l}$ or $M_{max}^{r}$ is the moment capacity of the relevant end section as long as that value is lower than the moment capacity of the structures that the end is connected to. Otherwise, $M_{max}^{l}$ and/or $M_{max}^{r}$ have to be taken equal to the moment capacity of the structures at the boundaries. Specifically, $M_{max}^{l}$ or $M_{max}^{r}$ is the moment capacity of the end section if the plastic hinge develops at that end of the beam. If, conversely, the plastic hinge develops in the structures at the boundary, $M_{max}^{l}$ or $M_{max}^{r}$ has to be taken equal to the ultimate bending moment of the actual plastic hinge (or both of them).

In many beams, the end sections have equal geometry and reinforcement. Consequently, $M_{max}^{l}$ = $M_{max}^{r}$ ≡ $M_{max}^{e}$. In that case, Equation (17) becomes
(19)$$q_{u}=\frac{8\cdot {\left(\delta \cdot M_{max}^{m}+M_{max}^{e}\right)}_{}}{{L^{2}}^{}},$$
in which δ is calculated using the *y*/*d* ratio of the end section.

Moreover, Equation (18) becomes
(20)$$q_{u}=\frac{8\cdot {\left(M_{max}^{m}+\delta \cdot M_{max}^{e}\right)}_{}}{{L^{2}}^{}},$$
in which δ is calculated using the *y*/*d* ratio of the midspan section.

The results of the non-linear analysis were also used to obtain a criterion that allows the last plastic hinge to be identified beforehand, so as to know which equation must be used, i.e., whether Equation (17) or Equation (18), or whether Equation (19) or Equation (20). The criterion is based on the following ratio:(21)$$\lambda =\frac{M_{max}^{m}}{M_{max}^{e}},$$
where $M_{max}^{e}$ is the greatest value between $M_{max}^{l}$ and $M_{max}^{r}$.

If λ ≤ 0.5 the plastic hinge that develops last is the one at the end which transmits $M_{max}^{e}$, while, if λ > 0.5, the plastic hinge that develops last is the one at midspan. Actually, if $M_{max}^{l}$ and $M_{max}^{r}$ differ a lot from each other and, simultaneously, λ is around 0.5, that condition can fail the prediction. In this case, either a deeper insight is gained or the formula that is used is the one providing the lowest ultimate load so as to be conservative.

Summarizing, the sequence of steps involved in moving from the beginning to the end of the calculus (i.e., the workflow) is as follows:-The first step involves Equation (14). That formula is used to determine the moment capacity at the midspan, left end, and right end. That is, $M_{max}^{m}$, $M_{max}^{l}$, and $M_{max}^{r}$ are derived from Equation (14).-The second step involves Equation (21). That formula is used to determine λ.-The third step involves Equation (16) and uses λ from the second step. That formula is used to determine δ. If λ ≤ 0.5, *y*/*d* to plug into Equation (16) is the ratio of the plastic hinge at midspan. If λ > 0.5, *y*/*d* to plug into Equation (16) is the ratio of the plastic hinges at the ends.-The fourth step involves either Equation (17) or Equation (18), and uses δ from the third step. If the end sections of the beam have equal geometry and reinforcement, those equations can be replaced by either Equation (19) or Equation (20). This step provides the ultimate load *q_u_* of the RC beam. If λ ≤ 0.5, δ has to be plugged into Equation (17) or Equation (19), and the formula provides *q_u_*. If λ > 0.5, δ has to be plugged into Equation (18) or Equation (20), and the formula provides *q_u_*.

## 10. Estimation of the Accuracy

In order to prove the accuracy of the proposed formulas, the values of δ from Equation (16) were compared to the values of δ that made Equations (16)–(19) provide the same results obtained from the non-linear analysis (Figure 6). Specifically, Figure 6 was constructed calculating the values of δ to plug into those equations in order to obtain the ultimate loads provided by the non-linear analysis.

That construction is the inverse of the process that Equation (16) was derived from, which is described in Section 6.

Ultimately, Figure 6 shows how well the curve described by Equation (16) fits the points provided by the non-linear analysis. In other words, the comparison between the values of δ derived from the non-linear analysis and the curve described by Equation (16) is a measure of the accuracy of Equations (17)–(20), since the former are the actual values of δ (checked against experimental results as well), while the latter are the values proposed in lieu of the former to use in Equations (17)–(20).

The moderate differences between the points and the curve prove that the equation proposed herein provides accurate estimations of the ultimate load. Moreover, according to this comparison, that equation is conservative (apart from unrealistically low ductilities, which imply very high *y*/*d* ratios).

## 11. Application

Equations (17)–(20) were applied to the structure shown in Figure 7—i.e., an RC beam supported by two columns. The purpose of this example is to evaluate the differences between the results from the proposed formula and from a plastic analysis, as well as to show how to apply the formula.

As shown by Figure 7, the end sections have the same steel reinforcement (both in tension and compression), and the midspan section has the same steel reinforcement as each end section (of course, at midspan, the tension reinforcement is at the bottom edge, while the compression reinforcement is at the top edge, while the opposite is true at the ends).

The design value of the concrete compressive strength (crushing stress) is 15.0 N/mm^2^, and the design value of steel strength (yielding stress) is 391 N/mm^2^.

According to Equation (21), λ = 1 (see Section 9). Thus, the last plastic hinge that develops is the one at midspan, while the first plastic hinge that develops is at the end. More specifically, the two plastic hinges at the ends theoretically form simultaneously. The *y*/*d* ratio at the end section results to be 95.67/200.0 = 0.478.

According to Equation (16), that ratio gives δ = 0.347. Plugging that value of δ into Equation (15), the plastic hinge at midspan results as 34.7% of its moment capacity. Specifically, the last plastic hinge, which is that at midspan, balances the ultimate load with only 34.7% of *M_max_* (i.e., 34.7% of the bending moment provided by Equation (14) for that section or by an equivalent expression, if preferred).

Put differently, when the plastic hinge at midspan reaches its moment capacity *M_max_*, the plastic hinges at the ends have already failed and transmit marginal bending moments, and consequently the load that the beam can balance is lower than the ultimate load (post-failure behavior).

According to Equation (14), *M_max_* = 170.1 kN⋅m. That value applies to the three plastic hinges, since those sections have the same steel reinforcement.

The ultimate load *q_u_* is predicted by Equation (19), since the end sections have equal geometry and reinforcement, and the plastic hinge that develops last is that at midspan.
(22)$$q_{u}=\frac{8\cdot {\left(0.347\cdot 170.1+170.1\right)}_{}}{6.0^{2}}=50.92\;\mathrm{kN}/m.$$

The plastic analysis would have provided *q_u_* = 75.60 kN⋅m, which is 75.60/50.92 = 1.48 times greater than the actual value, provided by Equation (22).

The beam that is analyzed in this section, shown in Figure 6, was tested in Reference [60]. The experimental load-carrying capacity that was found was 54.13 kN/m (the loads that were applied in those tests were not exactly uniform, since each loading was distributed onto 4/5 of the spans, i.e., the load distributions did not reach the ends of the beam).

## 12. Conclusions

This paper presents a research work devoted to predicting the bending load-carrying capacity of hyperstatically supported RC beams allowing for ductility, which dictates whether the failure mode is fully developed or premature and how much it is premature. The major finding is the formula that predicts the maximum load that may be placed on a beam before its collapse due to premature failure.

The expected research impact within academia is that this novel approach that blends plastic analysis and non-linear analysis will be considered by researchers engaged in investigating the ultimate behavior and strength of RC structures. The expected research impact beyond academia is that structural engineers will use this method to design and assess RC beams, and to determine to what extent an increase in the amount of longitudinal steel reinforcement counteracts a reduction in height of the concrete section.

The underlying theoretical support for this activity was a non-linear analysis, whose general outline and results were presented in the body of the paper. In more detail, the non-linear analysis was based on the constitutive law of concrete and steel, together with the moment–curvature relationship that allows for the tension stiffening effect in the cracked and post-yielding states. The non-linear analysis was conceived and developed so as to be precise without regard of complexity. Verification against experimental data confirmed the accuracy of the non-linear analysis results. Moreover, the non-linear analysis covered all reasonably possible practical applications. As such, the results from the non-linear analysis allowed general conclusions to be drawn, which gave the opportunity for applying some simplifications to the analytical formula that the research work aimed to obtain.

One of the results of the non-linear analysis is that the crushing strain of concrete plays a marginal role in the ultimate load, as long as its value is greater than 2.5–3.5‰, which is always verified apart from a very poor concrete.

Another result is that the effects of ductility on the ultimate load can be synthesized by the ratio of the neutral axis depth to effective depth. Consequently, the ultimate load can be predicted by the mechanism equation of the plastic analysis, applying a reduction coefficient to the plastic moment. This coefficient depends only on that ratio.

On the basis of those results, research activity calibrated a function against non-linear analysis results, in order to describe analytically the relationship between ultimate load and ductility. That function provides a coefficient, denoted by δ, which introduces the relationship between load-carrying capacity and rotational capacity, in the ultimate equilibrium equation of plastic analysis. Specifically, the coefficient δ provided by the function that was calibrated herein, converts the potential fully developed failure mode into the actual premature failure mode. Ultimately, δ transforms a load-carrying capacity existing only as an idea in the real load-carrying capacity.

That formula fits the non-linear analysis results well. Since the non-linear analysis results were checked against test results, for the transitive property, the proposed method provides accurate predictions of the ultimate load.

The formula that was proposed can also be applied if a plastic hinge develops beyond the boundary of the beam instead of at the end (or if two plastic hinges are beyond the boundaries). In this case, the equation has to use the moment capacity of each section that becomes a plastic hinge in lieu of the moment capacity of the beam end (the former must be lower than the latter; otherwise, the plastic hinge would not be beyond the boundary). Ultimately, if a column or an adjacent beam is weaker than the analyzed beam, the formula holds true.

The paper considered the uniform load. Nevertheless, the coefficient δ holds true for any load distributions, including concentrated forces. In order to apply the formula to loads different than the uniform load, the virtual work equation must be applied to the failure mechanism exhibited by the beams with the actual load distribution, and the reduction coefficient δ must be applied to the first hinge that develops.

In the case of beams with moderate (or low ductility), the differences between the ultimate load from plastic analysis and from non-linear analysis can be quite large (sometimes very large), since failure can be premature (sometimes drastically premature). Thus, the role played by ductility should not be ignored in the ultimate analysis. Those differences can be accounted for using non-linear analysis. However, non-linear analysis implies complex calculations, which are unsuitable for the design phase. In particular, they need a calculus code (i.e., finite element software), which can only be used when the design is completed and not during the design phase.

The formula that was presented in this paper, albeit an equilibrium equation, allows the role played by ductility in load-carrying capacity to be taken into account, since the rotational capacity is condensed into a coefficient. In so doing, accurate predictions of the ultimate load can be obtained by a mechanism equation, which is simple and does not need any software.

Ultimately, the proposed formula combines the straightforwardness of plastic analysis with the accuracy of non-linear analysis. Thus, it can be used not only for verification purposes, but also in the design phase. In particular, the author expects that the formula will be used by practitioners to know how much an increase in reinforcement can compensate for a reduction in beam height, as well as to identify premature failure.

## Figures and Tables

**Figure 1 materials-12-03085-f001:**
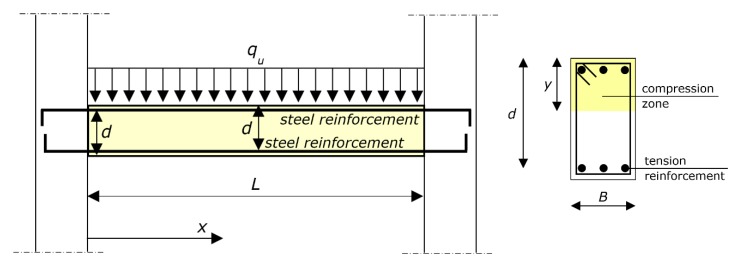
Reference structure. On the left: hyperstatically supported reinforced concrete (RC) beam (in the schematic, the beam is connected to vertical structures, e.g., columns or walls), and ultimate load *q_u_*. The end sections are at *x* = 0 and *x* = *L* (span). The effective depth is denoted by *d*. On the right: cross-section of the beam at midspan. The width is denoted by *B*. The neutral axis depth to effective depth ratio is denoted by *y*/*d*. The stirrups, which are not represented in the left diagram, have no role in the flexural behavior, since the concrete is unconfined or lightly confined.

**Figure 2 materials-12-03085-f002:**
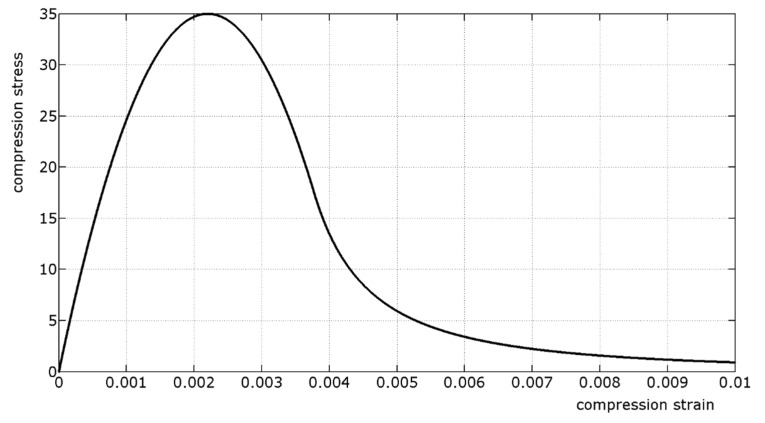
σ*c*–ε*c* relationship for uniaxial compression of concrete, described by Equations (2)–(7).

**Figure 3 materials-12-03085-f003:**
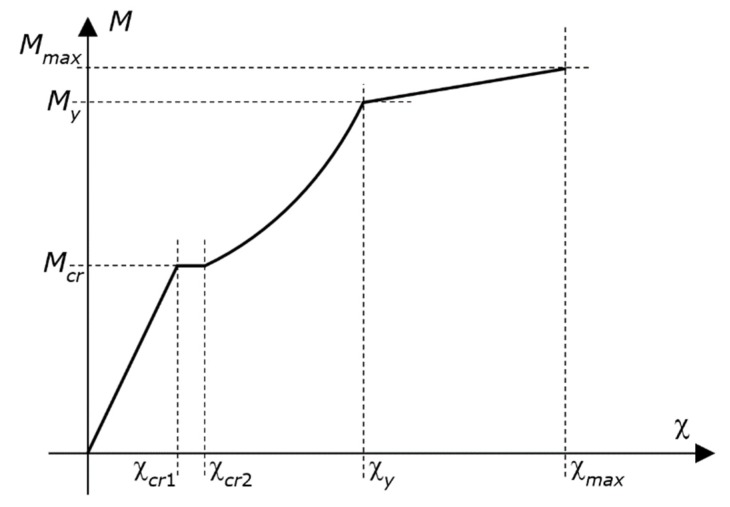
Diagram of the *M*–χ relationship described by Equations (8)–(10). The figure also shows the symbols.

**Figure 4 materials-12-03085-f004:**
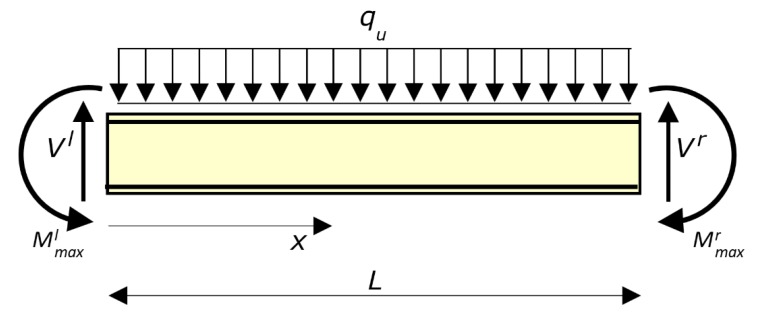
Second step of the method for calculating the ultimate load *q_u_*. The bending moments $M_{max}^{l}$ and $M_{max}^{r}$ are the moment capacity of the left end and of the right end, respectively (or of the adjoining structures, if weaker). The shear forces *V^l^* and *V^r^* result from equilibrium.

**Figure 5 materials-12-03085-f005:**
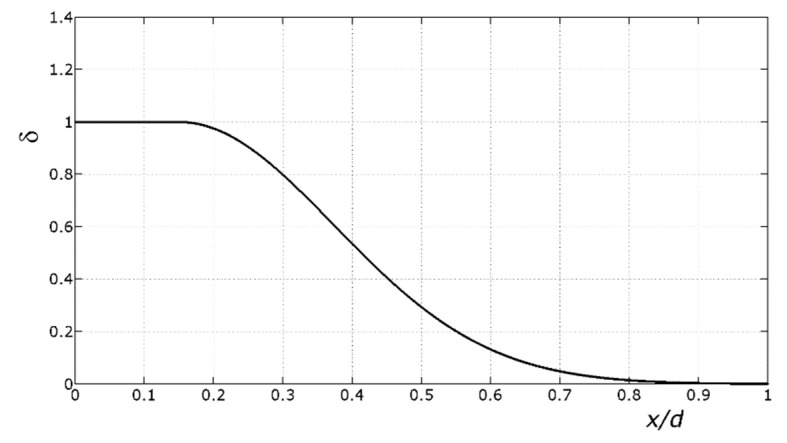
Graphic of Equation (16), which relates the coefficient δ to the ratio of the neutral axis depth to effective depth *y*/*d* of the first plastic hinge that forms. The coefficient δ reduces the bending moment of the last plastic hinge(s).

**Figure 6 materials-12-03085-f006:**
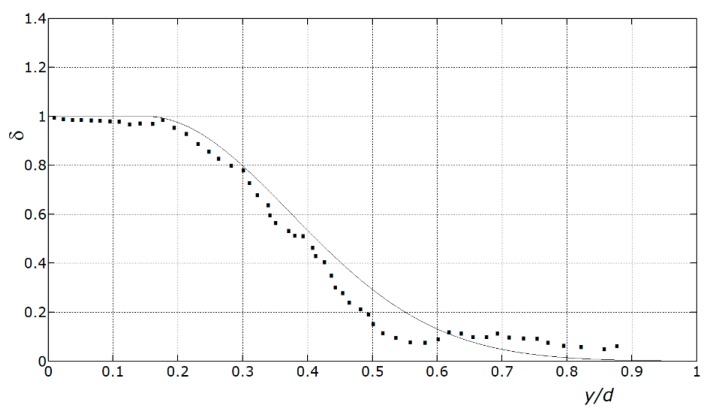
Comparison between the results from the proposed formula and the results from the non-linear analysis. Solid line: δ obtained from Equation (15). Points: values of δ to plug into Equations (16)–(19) in order to obtain the ultimate load provided by the non-linear analysis.

**Figure 7 materials-12-03085-f007:**
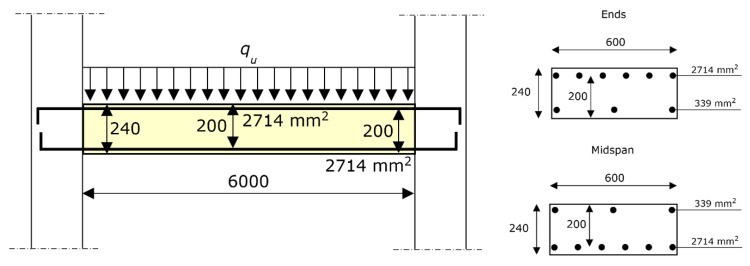
RC beam considered as an applicative example. On the **left**: longitudinal section. On the **right**: cross-sections at the ends (above) and at the midspan (below). The dimensions are expressed in millimeters. The figure also shows the amount of reinforcement at the ends and at the midspan, which are equal.

**Table 1 materials-12-03085-t001:** Bending moment transmitted by the last plastic hinge, *M’_u_* = δ⋅*M_max_*, for different steel reinforcement amounts, sections, and concrete grades. The table reports the ratio $\Phi =\frac{{\delta \cdot M}_{max}}{{B\cdot d\cdot f}_{cm}}$ and the ratio $\omega =\frac{A_{s}\cdot f_{sd}}{{B\cdot d\cdot f}_{cm}}$. The symbols Φ*_e_* and Φ*_t_* denote the experimental and theoretical values of Φ, respectively. The symbol *d* 1s defined in Section 3, i.e., it denotes the effective depth of the cross-section. The upper row reports the value of ω. The two rows below report Φ*_e_* and Φ*_t_*.

	**0.104**	**0.158**	**0.163**	**0.201**	**0.257**	**0.295**	**0.354**	**0.389**	**0.401**
Φ*_e_*	0.9768	0.1436	0.1469	0.1770	0.2185	0.2441	0.2808	0.3004	0.3066
Φ*_t_*	0.9565	0.1390	0.1406	0.1676	0.2049	0.2264	0.2577	0.2726	0.2722

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
