# Peer review of "Bending Load-Carrying Capacity of Reinforced Concrete Beams Subjected to Premature Failure"

_materials, 2019, doi:10.3390/ma12193085_

Round 1

Reviewer 1 Report

This paper investigated bending load-carrying capacity of reinforced concrete beams subjected to premature failure and proposed a formula. The following comments are suggested:

The contents of the first chapter and the second chapter are too lengthy and should be described in a concise manner for the reader to read. In the title of Figure 1, "On the left" should be "above" and "On the right" should be "below". The stress-strain relationship of concrete is critical. Are Equations (2-7) self-derived or are they cited in the literature? If they are self-derived, please provide relevant test and analysis data. In a cracked RC member, the section of the un-cracked area has the maximum bending stiffness, and the cracked section at the crack has the least bending stiffness. However, in the section between two adjacent cracks, the concrete around the steel bar can still bear part of the tensile stress due to bond transfer. Due to the contribution of the concrete between the cracks to the tensile force, the tensile stress of the steel bars is lowered, and the neutral axis of the section is moved toward the tension zone. Therefore, the bending stiffness of the member between adjacent cracks will be higher than that of the crack, and the overall bending stiffness of the member will increase. This phenomenon is generally called the tensile stiffening effect or the tension stiffening effect. In view of this, the author considers the tension stiffening effect and corrects the moment-curvature relationship. Are Equations (8-10) self-derived or are they cited in the literature? If they are self-derived, please provide relevant test and analysis data. The literature cited in Equation (11) should be added. Page 9, lines 293-294, "The diagram of the M-c relationship described by Eqs. (8-10) is well-known. The curve is composed of three branches...". Please draw this diagram. The literature cited in Equation (12) should be added. In the title of Table 1, coefficient d appears for the first time and its definition should be stated. On page 13, Equation (12) should be Equation (13). By analogy, the numbers of the remaining equations must be modified. The literature cited in the revised Equation (13) should be added. In accordance with the modification of the equation number, the number of the equation cited in the text should also be adjusted. Overall, the contents of Chapters 8 and 9 can also be streamlined for easy reading. Please provide the flow chart of the analysis and calculation steps. In the article, the titles of the figure and table should be concise. A description of some of the symbols can be described in the text. Conclusions should summarize the important results of the research and should be streamlined.

Author Response

The contents of the first chapter and the second chapter are too lengthy and should be described in a concise manner for the reader to read.

The revised version resubmitted has added what suggested by the Reviewers and then has rewritten the article reducing the length as much as possible.

In the title of Figure 1, "On the left" should be "above" and "On the right" should be "below".

In the revised version resubmitted the two parts of figure 1 have been put into a table (with invisible rows and lines) so as to arrange them in a single row. On the contrary, in the manuscript the two parts moved and the second shifted on the row below.

The stress-strain relationship of concrete is critical. Are Equations (2-7) self-derived or are they cited in the literature? If they are self-derived, please provide relevant test and analysis data.

The revised version resubmitted has added the relevant literature where those formulas were borrowed from.

In a cracked RC member, the section of the un-cracked area has the maximum bending stiffness, and the cracked section at the crack has the least bending stiffness. However, in the section between two adjacent cracks, the concrete around the steel bar can still bear part of the tensile stress due to bond transfer. Due to the contribution of the concrete between the cracks to the tensile force, the tensile stress of the steel bars is lowered, and the neutral axis of the section is moved toward the tension zone. Therefore, the bending stiffness of the member between adjacent cracks will be higher than that of the crack, and the overall bending stiffness of the member will increase. This phenomenon is generally called the tensile stiffening effect or the tension stiffening effect. In view of this, the author considers the tension stiffening effect and corrects the moment-curvature relationship. Are Equations (8-10) self-derived or are they cited in the literature? If they are self-derived, please provide relevant test and analysis data.

The revised version resubmitted has added the relevant literature where the equation of the moment-curvature relationship were borrowed from.

The literature cited in Equation (11) should be added.

The revised version resubmitted has added the literature relevant to equation (11).

Page 9, lines 293-294, "The diagram of the M-c relationship described by Eqs. (8-10) is well-known. The curve is composed of three branches...". Please draw this diagram.

The revised version resubmitted has included the diagram of the M-C relationship described by Eqs. (8-10).

The literature cited in Equation (12) should be added.

The revised version resubmitted has provided explanation about equation (12).

In the title of Table 1, coefficient d appears for the first time and its definition should be stated.

In the revised version resubmitted, the symbol d is defined in Section 3 about the reference structure and is repeated in the caption of Table 1.

On page 13, Equation (12) should be Equation (13). By analogy, the numbers of the remaining equations must be modified.

Corrected. Thank you very much.

The literature cited in the revised Equation (13) should be added.

The revised version resubmitted has provided explanation about equation (13).

In accordance with the modification of the equation number, the number of the equation cited in the text should also be adjusted.

Corrected. Thank you very much.

Overall, the contents of Chapters 8 and 9 can also be streamlined for easy reading. Please provide the flow chart of the analysis and calculation steps.

The revised version resubmitted has added what suggested by the Reviewers. In particular, the process has been summarized in a flow chart expressed as a workflow. Then the revised version has rewritten chapters 8 and 9 reducing the length as much as possible.

In the article, the titles of the figure and table should be concise.

Corrected.

A description of some of the symbols can be described in the text.

The revised version resubmitted has added the definition of the symbols in the text.

Conclusions should summarize the important results of the research and should be streamlined.

The revised version resubmitted has made the conclusion more readable.

Reviewer 2 Report

There are some weaknesses through the manuscript which need improvement. Therefore, the submitted manuscript cannot be accepted for publication in this form, but it has a chance of acceptance after revise and resubmit. My comments and suggestions are as follows:

1- Abstract gives information on the main feature of the study. Some more details about type of the studied concrete should be added.

2- In introduction, there are several papers cited in the manuscript, but they are not reviewed. For instance: Ref. [1-13] and Ref. [14-20]. It is better to comment each reference at least in couple of sentences.

3- Although a main aim of the presented manuscript is mentioned in section 2.2 (page 4), it seems better to explain aims and objective of the study in last part of the introduction.

4- It is necessary to mention the main source (reference) of the utilized formulas. Moreover, each parameter must be introduced.

5- Since in the submitted manuscript considered nonlinear analysis, it is recommend reading and citing following studies which discussed nonlinear analysis:
- Engineering Structures, 190:116-127 (2019)
- Journal of Theoretical and Applied Physics, 10:211-218 (2016)
- Journal of Constructional Steel Research, 106:209-219 (2015)

6- Since experimental verification is presented in Section 7, it would be better if figure(s) added to this section to illustrated experimental practice.

7- It would be better if the descriptions are more concise. I believe the sections can be compacted and reduced with no detriment for the manuscript.

Author Response

1- Abstract gives information on the main feature of the study. Some more details about type of the studied concrete should be added.

The abstract of the revised version resubmitted has added details about the type of concrete that were studied.

2- In introduction, there are several papers cited in the manuscript, but they are not reviewed. For instance: Ref. [1-13] and Ref. [14-20]. It is better to comment each reference at least in couple of sentences.

In the revised version the citations are associated to the specific topic each one is related to. 

3- Although a main aim of the presented manuscript is mentioned in section 2.2 (page 4), it seems better to explain aims and objective of the study in last part of the introduction.

Corrected. 

4- It is necessary to mention the main source (reference) of the utilized formulas. Moreover, each parameter must be introduced.

Corrected. The revised version resubmitted provides the reference each formulas was borrowed from .

5- Since in the submitted manuscript considered nonlinear analysis, it is recommend reading and citing following studies which discussed nonlinear analysis

The revised version resubmitted cited the following papers.

[10] T. Lou, T.L. Karavasilis. Numerical assessment of the nonlinear behavior of continuous prestressed steel-concrete composite beams. Engineering Structures, 2019; 190(July): 116-127.

[22] T. Limazie, S. Chen Numerical procedure for nonlinear behavior analysis of composite slim floor beams. Journal of Constructional Steel Research, 2015; 106(March): 209-219.

[40] P. Valipour, S.E. Ghasemi, M.R. Khosravani, D.D. Ganji. Theoretical analysis on nonlinear vibration of fluid flow in single-walled carbon nanotube. Journal of Theoretical and Applied Physics, 2016. 10(3), September: 211–218.

6- Since experimental verification is presented in Section 7, it would be better if figure(s) added to this section to illustrated experimental practice.

The revised version resubmitted cited the paper that presented the research. All the photos of the experiments are included in that paper, so I could not include those photo due to the copyright.

7- It would be better if the descriptions are more concise. I believe the sections can be compacted and reduced with no detriment for the manuscript.

Corrected. A more concise description is presented in the revised version resubmitted.